# Beyond Magnitude and Gradient: Network Pruning Inspired by Optimization Trajectories

## Abstract

Deep neural networks are dramatically over-parameterized and can be pruned without effecting the generalization. Existing pruning criteria inspect weights or gradients in isolation and ignore the effect of optimization dynamics on pruning. We introduce Causal Pruning (CP) – A method by which one learns the parameter-importance from the *optimization trajectory* directly.

We exploit the "causal" signal hidden in SGD trajectories, where each weight update is considered as an intervention and measuring its effect on the loss – observed versus predicted. This view yields two insights: (i) a weight's importance is proportional to the gap between the predicted loss change (via a first-order Taylor estimate) and the observed loss change, and (ii) at convergence, weights whose removal leaves the local basin no sharper – i.e. does not reduce flatness – can be pruned without harming generalization. Empirically, we show that causal pruning is comparable to recent state-of-the-art approaches.

## 1 Introduction

Gradient descent (GD) is one of the important reasons why deep neural networks works well (Arora et al., 2019). Primarily, the high-dimensional nature of the loss-landscape allows gradient descent to obtain solutions which generalize well. Arora et al. (2019) argues that the redundancy due to the width and depth in the networks allows GD to overcome the non-convexity issues. On the flip side, while redundancy is important for finding the solution, it is not required for inference and can easily be removed or *pruned* for efficiency. In a very different line of research, several techniques have been developed for pruning neural networks (Kalchbrenner et al., 2018; Hoefler et al., 2021). In this article we ask -

> *Does the inherent redundancy in stochastic gradient descent yield parameters that can be pruned without compromising the network's performance? If so, how can one identify them?*

The key idea here is that – Gradient descent implicitly (and inefficiently) performs causal reasoning[1]. And, by making the causal relationship explicit we obtain a simple and theoretically founded heuristic for pruning. We refer to this procedure as *causal pruning.*

**Implicit causality in gradient descent:** Note that every step of gradient descent results in a change of both the parameters $\theta$ and the loss function $L$. The change in parameters is dictated by the gradient of the loss function $\partial L/\partial \theta$. The implicit causality within this model is that - Changing $\theta \to \theta + \Delta\theta$ is expected to change the loss $L \to L + \Delta L$, where $\Delta L$ and $\Delta\theta$ are related by the first-order gradient information. In other words, *"changing the parameters $\theta$ causes the change in the loss"*. This however is not always true - there usually are parameters which do not result in the reduction in the loss. Thus, the implicit causality in the gradient descent is not perfect. We make the causality relationship explicit in section 4.

---

[1]Throughout the article we use the word causality to refer to Granger-type causality and not causality in the sense of graphical models.

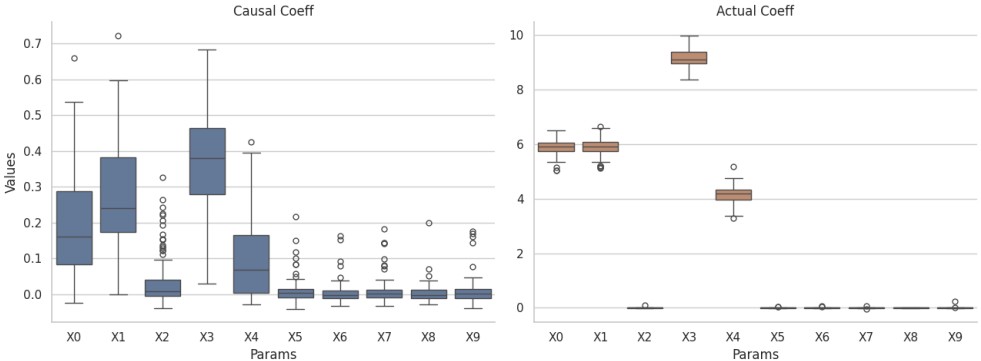

Figure 1: Proof of concept that causal coefficients from SGD recovers feature importance on a synthetic Friedman benchmark.The ground truth generative process uses five features $X0$-$X4$ with varying importance and non-linear interactions, while five features $X5$-$X9$ are irrelevant. Right panel shows the distribution of coefficients from a Lasso regression, estimated over 1000 data instantiations, identifies the zero-importance features and captures the relative importance of the others. Left panel shows that the boxplot of causal coefficients from SGD, and it shows a nearly identical relative ordering. This confirms that our method effectively recovers the same feature attributions as the well-established Lasso on this controlled problem.

**Proof of Concept:** Figure 1 presents a simple illustration of our approach. We use the standard Friedman dataset(Friedman, 1991; Breiman, 1996), which includes both informative and uninformative features. Specifically, features X0–X4 carry signal with varying levels of importance, while features X5–X10 are entirely irrelevant.

The ground-truth feature importance is obtained using lasso regression. The resulting importances appear in the right panel of Figure 1. Next, we use the gradient trajectory to compute feature contributions via the causal model – $\Delta L = \sum_k \gamma_k (\Delta \theta)^2$ (see Section 3 for derivation) – where $\Delta L$ is the actual change in the loss and $(\Delta \theta)$ is the change in the parameters due to gradient descent. This formulation captures how changes in parameters influence the loss during training. The comparison in Figure 1 shows a close match between the importances estimated by lasso and those derived from the causal model. This alignment confirms that gradient trajectories encode meaningful information about feature relevance.

We leverage this insight throughout the paper to identify which parameters are critical and which can be safely pruned. Our findings suggest that SGD trajectories offer a reliable signal for assessing parameter importance.

## Summary of Key Results/Contributions.

*1. Causal importance metric.* We reinterpret each stochastic gradient update as a causal intervention on the loss. This perspective allows us to derive a per-parameter coefficient, $\gamma_k$, that measures the *discrepancy* between the predicted and observed change in loss. Unlike traditional pruning scores based on parameter magnitude (Han et al., 2016) or Hessian curvature (Singh & Alistarh, 2020), our criterion learns directly from the optimization trajectory.

*2. Causal Pruning algorithm.* We introduce a novel pruning method based on lasso regression over SGD traces. The regression ranks parameters by their $\gamma_k$ values and prunes those with minimal causal impact. Our algorithm integrates seamlessly with any architecture, adds negligible overhead, and avoids the need for second-order derivatives or Hessian computations. Figure 2b illustrates the distinct behavior of causal pruning compared to traditional magnitude-based approaches. Rather than removing weights solely based on their magnitude, causal pruning identifies and eliminates parameters that contribute least to actual loss reduction.

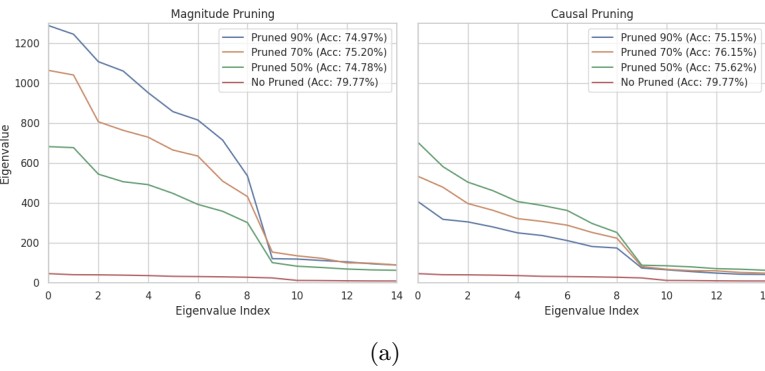 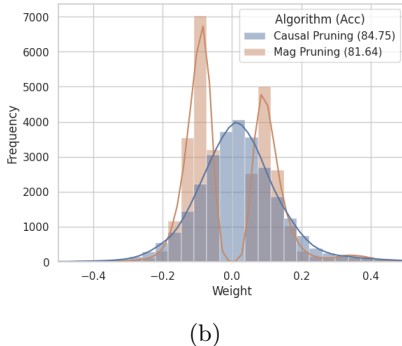

(a)          (b)

Figure 2: (a) Top eigenvalues of LeNet trained on CIFAR-10 under both magnitude and causal pruning. Observe that (i) causal pruning yields significantly fewer dominant eigenvalues, and (ii) unlike magnitude pruning, the eigenvalues *decrease* with more pruning, indicating a reversal in spectral behavior. (b) Histogram of remaining weights in ResNet-20 on CIFAR-10 after 90% pruning. Observe that (i) magnitude pruning largely eliminates near-zero weights, while causal pruning targets a markedly different subset, and (ii) without any finetuning, causal pruning achieves higher accuracy (84%) than magnitude pruning (81%), suggesting it prunes both large and small weights in a complementary manner.

*3, Theoretical Analysis.* We show that pruning parameters with $\gamma_k \neq 0$ preserves the descent direction during early optimization and maintains the dominant eigenvalues of the Hessian at convergence. These results connect our criterion to the flat-minima hypothesis and highlight its stability-aware behavior.

*4. Empirical evidence.* Causal Pruning achieves performance competitive with state-of-the-art methods across a range of architectures and vision benchmarks (See section 5 for details). Moreover, the pruned models exhibit significantly flatter Hessian spectra. Figure 2a reveals this behavior. We plot the top eigenvalues of the Hessian for LeNet trained on CIFAR-10 under both causal and magnitude pruning and observe: (i) causal pruning produces significantly smaller top-eigenvalues, and (ii) in contrast to magnitude pruning, the spectral norm decreases with increased sparsity, indicating a reversal in the Hessian's spectral profile.

## 2 Literature Review

Surprisingly, while many pruning methods leverage gradient or Hessian information to estimate parameter importance, they typically ignore the role of the optimization dynamics. To our knowledge, no prior work explicitly incorporates the influence of gradient updates, as realized through the steps of gradient descent, into the pruning criterion. Below we survey some related aspects to our work - pruning, gradient based methods

**Related Literature on Pruning:** Pruning neural networks has gathered a lot of attention for variety of reasons - (i) Improves the efficiency of models at inference without compromising the accuracy (Kalchbrenner et al., 2018; Hoefler et al., 2021), (ii) Lottery ticket hypothesis (LTH) (Frankle & Carbin, 2019; Jin et al., 2022; Paul et al., 2023) and related works show that pruning experiments can help us understand the working of optimization and generalization in the training of neural networks.

*Magnitude Based Pruning:* The popular heuristic that, the magnitude of the parameter reflects the importance of the weight, underlies several existing pruning methods. (Han et al., 2016; Gordon et al., 2020; Wang et al., 2023). In fact, as Wang et al. (2023) notes, using magnitude-based pruning at the level of filters (a.k.a $L_1$ norm pruning), the authors could achieve state-of-the-art results with this simple heuristic.

*Gradient information based pruning:* Tran et al. (2022) shows that small gradient norms contribute less to the loss and hence provides a justification for pruning. Redman et al. (2022) uses gradient magnitudes as a scoring function for pruning. Fladmark et al. (2023) uses fisher information based on gradients for scoring function. Zhang et al. (2022) integrates pruning directly into the gradient updates, where one thresholds

---

**Algorithm 1** Causal Pruning

---

1: **Parameters:** $N_{pre}$, $N_{iter}$, $N_{prune}$, $N_{post}$, $p$.
2: **Input:** Model:$f_\theta$, Dataset:$\{\boldsymbol{x}_i, y_i\}$
3: Train the model for $N_{pre}$ number of epochs.
4: **for** $i = 1, 2, \cdots, N_{iter}$ **do**
5:    Train the model for $N_{prune}$ epochs and collect the values of the parameters and the losses after each gradient step. Let $\{(\theta^t, L^t)\}$ denote the values after each gradient step.
6:    Using `Lasso Regression`, fit the model in equation 3 using the $L_1$ coefficient `L1_coeff`. Prune the smallest $p$ fraction of $\{\gamma_k\}$
7:    Reset the remaining weights to be the ones after step 3.
8: **end for**
9: Complete the training of the model for $N_{post}$ number of parameters.

---

the parameters based on an importance score and sensitivity. This is very similar to using $L_1$ regularization for pruning (Buschjäger & Morik, 2023). Gradient based methods for pruning have also been tested for very large networks such as llms (Das et al., 2023).

*Impact Based Pruning:* Several pruning methods (LeCun et al., 1989; Singh & Alistarh, 2020) measure the dip in the loss function with respect to the parameters and decide which parameters to prune. Approaches such as one proposed by Singh & Alistarh (2020) use second-order Taylor approximation for the criterion. Benbaki et al. (2023) uses combinatorial optimization and approximates the Hessian using a low-rank matrix.

*Remark:* Please note that we do not aim for an exhaustive survey. Instead, we highlight representative ideas from the literature to contrast with our approach.

## 3 Causal Pruning Algorithm

**Notation:** Let $p_{\text{data}} = \{\boldsymbol{x}_i, y_i\}$ denote the dataset. Let $f_\theta$ denote the network to be trained. Let $L(\theta)$ denote the loss function used to optimize $\theta$ using Stochastic Gradient Descent (SGD). We use $\partial L(\theta^t)/\partial \theta$ to denote the derivative of $L$ with respect to $\theta$ at $\theta = \theta^t$. Let $\theta^0, \theta^1, \cdots, \theta^t, \cdots$ denote the path in the parameter space taken by the gradient descent. Further, we let

$$(\Delta L)^t = L(\theta^t) - L(\theta^{t-1}) \tag{1}$$

Let $\theta_i^t$ denote the $i^{th}$ parameter in the vector $\theta^t$. Then, we denote,

$$(\Delta \theta_k)^t = \theta_k^t - \theta_k^{t-1} \tag{2}$$

**Causal Pruning Algorithm.** Consider gradient descent iterations indexed by $t \in \{1, \ldots, T\}$, with observed parameter updates $\{(\Delta \theta_k)^t\}_{t=1}^T$ and corresponding loss variations $\{(\Delta L)^t\}_{t=1}^T$. We introduce a sparsity-promoting regression framework to identify causal parameters by solving the following optimization problem:

$$\sum_t \left( \Delta L^t - \sum_k \gamma_k (\Delta \theta_k^t)^2 \right)^2 + \alpha \sum_k |\gamma_k| \tag{3}$$

where $\gamma_k$ represent the learned causal importance weights and $\alpha$ controls the sparsity of the solution. Notably, we leverage the parameter updates and loss changes inherently generated during gradient descent itself, thus requiring no additional sampling effort beyond the natural optimization trajectory.

**Remark (Pre and Post training):** Given a dataset and architecture, we train the model for $N_{pre}$ epochs. This is crucial as pointed out by Blalock et al. (2020). This phase decides the *basin of attraction* of the parameters after which the model can be pruned more effectively. To maximize the performance of the network, we train it further (using only unpruned weights) for $N_{post}$ number of epochs.

**Optimization of equation 3:** For some networks, the number of parameters within each layer can go up to $16M$ and hence can cause memory issues. Shalev-Shwartz & Tewari (2011); Tsuruoka et al. (2009) discusses several algorithms for stochastic optimization of Lasso models as in equation 3. We consider a different approach in this article.

Note that, solving equation 3 with a given $L1$ coefficient ($\alpha$) removes an *arbitrary* number of parameters. However, for the purpose of pruning, one needs more control over the number of parameters we prune. To achieve this, we select the smallest $p$ fraction of $\{\gamma_k\}$ to remove instead.

**Computational Complexity:** Apart from the usual training of the networks, this procedure requires two additional steps - (i) Saving the checkpoints of the model after each gradient step for a few epochs, and (ii) Solving the lasso regression of the equation 3. Note that saving the checkpoints does not increase the computational complexity, but does require a larger storage. Further, these checkpoints can be removed after the computation of the coefficients in equation 3. Solving equation 3 is a straightforward problem and there exist several efficient solutions which use stochastic gradients (Shalev-Shwartz & Tewari, 2011; Tsuruoka et al., 2009). The naive approach has the computational complexity of $\mathcal{O}(mdk)$ where $m$ is the number of samples, $d$ is the number of features and $k$ is the number of epochs required to reach the solution. Note that it scales linearly in both the number of samples and some parameters (features), and hence very efficient. Further, these algorithms can also be parallelized, leading to higher gains.

**Parameter-Level or Layer-Level or Entire-Network:** It sometimes helps to prune channels or layers instead of individual parameters. This is referred to as structured pruning (He & Xiao, 2024). One can adapt the procedure in equation 3 to structured pruning using the strategy of *weight-sharing*. In essence, enforce the constraint that $\gamma_k$ is equal for all parameters $\theta_k$ within a layer. In this article, we only consider Parameter-Level pruning a.k.a unstructured pruning.

**Collecting the data for fitting equation 3:** Note that, one has to the collect the $\Delta L$ and $\Delta \theta$ to perform the causal pruning in equation 3. Since we are interested in measuring the effect of $\theta$ on the loss – *We collect the loss before and after the gradient step to compute $\Delta L$.*

## 4 Analysis of causal pruning

### 4.1 Why is our algorithm called causal pruning?

The main intuition behind causal pruning draws from Granger-Causality (Granger, 1969). We emphasize that our goal is to present the conceptual foundation underlying causal pruning rather than directly applying Granger-Causality in its traditional form.

**Vanilla Gradient Descent:** Firstly, we suitably modify the standard gradient descent as follows. Recall that the parameter update of gradient descent is given by

$$\theta^{t+1} = \theta^t - \eta \frac{\partial L(\theta^t)}{\partial \theta} \tag{4}$$

where $\eta$ denotes a fixed learning rate. Now, using a first-order Taylor approximation of $L$, we have

$$L(\theta^{t+1}) = L(\theta^t - \eta \frac{\partial L(\theta^t)}{\partial \theta}) \approx L(\theta^t) - \eta \left( \frac{\partial L(\theta^t)}{\partial \theta} \right)^T \frac{\partial L(\theta^t)}{\partial \theta} \tag{5}$$

Substituting from equation 4,

$$L(\theta^{t+1}) \approx L(\theta^t) - \eta \left( \frac{\theta^t - \theta^{t+1}}{\eta} \right)^T \left( \frac{\theta^t - \theta^{t+1}}{\eta} \right) = L(\theta^t) - \frac{1}{\eta} \|\theta^t - \theta^{t+1}\|^2$$
$$= L(\theta^t) - \frac{1}{\eta} \sum_k (\theta_k^t - \theta_k^{t+1})^2 \tag{6}$$

**Recall Granger Causality:** The Granger Causality test proposed in Granger (1969) is a classic test used to predict future values of a time series by using another time series. If $X_t$ and $Y_t$ denotes two time-series, we check if $X_t$ can forecast $Y_t$ by performing two (auto-)regressions as follows,

$$
\begin{aligned}
(R1) \quad Y_t &= a_0 + a_1 Y_{t-1} + a_2 Y_{t-2} + \cdots + a_2 Y_{t-m} + \text{error}_t \\
(R2) \quad Y_t &= a_0 + a_1 Y_{t-1} + a_2 Y_{t-2} + \cdots + a_2 Y_{t-m} + b_p X_{t-p} + \cdots + b_q X_{t-q} + \text{error}_t
\end{aligned}
\tag{7}
$$

One can interpret that $X_t$ can forecast $Y_t$ if the error under regression (R2) is smaller under (R1). The same can also be extended to multivariate analysis.

**Connecting Granger Causality with Gradient Descent:** The main idea is to observe that, if we take $Y_t = L(\theta^t)$ and $X_{t,k} = (\theta_k^{t+1} - \theta_k^t)^2 = ((\Delta \theta_k)^{t+1})^2$, equation 6 becomes,

$$
Y_{t+1} = Y_t + \sum_k \gamma_k X_{t,k},
\tag{8}
$$

where $\gamma_k = -1/\eta$. That is, $\gamma_k$ is a *constant independent of $k$*. Explicitly, the granger causal interpretation of gradient descent becomes,

$$
\text{Restricted: } Y_{t+1} = Y_t, \quad \text{Unrestricted: } Y_{t+1} = Y_t + \sum_k \gamma_k X_{t,k}.
\tag{9}
$$

If the unrestricted model provides a statistically significant improvement in the prediction accuracy we can establish that $X_{t,k}$ Granger-causes $Y_t$.

**Argument that Gradient descent has implicit causality:** From above, it is clear the gradient descent assumes a causality in its dynamics. In words, equation 6 reveals an implicit causal model within GD, albeit an approximate one. It suggests that the loss reduction is 'caused' by the squared changes in each parameter, with every parameter having the same causal influence coefficient: $-1/\eta$. This uniform influence stems directly from the use of a single learning rate and the first-order approximation, which ignores the true curvature and complexity of the loss landscape.

**From Causality to Pruning via Sparsity:** This explicit causal model above provides a natural framework for network pruning. Existing network pruning criteria involve either assessing the sensitivity of the loss function to a weight's removal or evaluating the magnitude of the weight itself. We instead utilize observational values to decide usefulness. If a parameter $\theta_k$'s changes, $(\Delta \theta_k)^2$ do not consistently contribute to the observed loss reduction $\Delta L$, its learned causal coefficient $\gamma_k$ should be close to zero. Such parameters are effectively "causally unimportant" within the recent training dynamics and can be pruned.

Estimating the coefficients $\{\gamma_k\}$ becomes a linear regression problem. Given data collected over $N_{prune}$ steps - pairs of $(\Delta L^t, \{(\Delta \theta_k^t)^2\}_k)$ - the goal is to find the $\gamma_k$ that best fit the model $\Delta L^t \approx \sum_k \gamma_k (\Delta \theta_k^t)^2$.

To encourage sparsity (i.e., force many $\gamma_k$ to be exactly zero), we employ Lasso regression (L1 regularization) to get the optimization problem in equation 3.

The L1 penalty term $\alpha \sum_k |\gamma_k|$ drives coefficients of features (here, $(\Delta \theta_k)^2$) with weak explanatory power towards zero. The pruning decision is then straightforward: prune parameter $\theta_k$ if its corresponding learned coefficient $\gamma_k$ is zero after solving the Lasso problem equation 3. The use of cumulative penalties over iterations is also relevant here (Tsuruoka et al., 2009). The approach by Tsuruoka et al. (2009) introduces an efficient SGD training method for L1-regularized log-linear models, particularly suited for high-dimensional sparse feature spaces. It achieves efficiency primarily through lazy updates, where parameters are only updated when their corresponding feature appears in the current training instance. Crucially, this update mechanism naturally allows parameters to become exactly zero: if the application of the cumulative penalty causes a parameter's value to cross zero, it is explicitly clipped to zero, thus enforcing the desired sparsity induced by L1 regularization. In addition, it forces the weight to receive the total L1 penalty that would have been applied if the weight had been updated by the true gradients.

**Remark:** Note that we have only considered vanilla gradient descent above. Extending this analysis to SGD+momentum approaches and possibly SGD+momentum+adaptive-learning rates will result in a more comprehensive granger-casual model. A brief discussion of this can be found in appendix A. In this article, we are only interested in the model for vanilla gradient descent.

**Remark:** Granger causality requires stationarity, which may not hold throughout entire training runs. To address this limitation, we employ short training windows with periodic weight resets for data collection – a standard practice in the pruning literature.

### 4.2 Characterizing the parameters which are pruned using Causal Pruning

Here, we try to analyze the causal pruning algorithm under different conditions, to obtain an intuition behind it's working. There are two phases one can identify during gradient descent (Ziyin & Ueda, 2023) - (i) An initial where where the first order relation control the dynamics and (ii) The second phase where one reaches an local optima and the second order relation control the dynamics. Causal pruning procedure acts differently in these two phases and we analyze the behavior under these phases below.

**First order analysis:**

Recall that for $t = 0 \ldots T$, we have the parameters $\{\theta^t\}$ and the corresponding losses $L(\theta^t)$. Causal pruning asks to fit the following (ignoring regularization),

$$\Delta L = \sum_k \gamma_k (\Delta \theta_k)^2 \tag{10}$$

However, a first order approximation of the loss $L$ gives $\Delta L \approx \sum_k \nabla_k L(\theta) \Delta \theta_k$, where $\nabla_k L(\theta) = \partial L(\theta)/\partial \theta_k$, which when applied at each time step $t$ gives

$$\Delta L^t \approx \sum_k \nabla_k L(\theta^t) \Delta \theta_k^t \tag{11}$$

In simple words, equation 11 denotes the approximate relationship between the data, and equation 10 denotes the relation we are trying to fit, i.e learn $\gamma_k$. From this, it is easy to see that

$$\gamma_k = \arg\min \sum_{t=1}^{T} (\nabla_k L(\theta^t) - \gamma_k \Delta \theta_k^t)^2 = \frac{\sum_{t=1}^{T} \nabla_k L(\theta^t) \Delta \theta_k^t}{\sum_{t=1}^{T} (\Delta \theta_k^t)^2} \tag{12}$$

**Important Remark:** Note that, $\Delta L$ denotes the change in the actual total loss and $\Delta \theta$ denotes the change in parameters. For *full gradient descent* we would have $\Delta \theta_k^t = \eta \nabla_k L(\theta^t)$. However, in stochastic gradient descent $\Delta \theta_k^t \neq \eta \nabla_k L(\theta^t)$ because the change in parameters is dictated by the *minibatch gradient descent*. Here we consider the latter case.

**Key Insight:** From equation 12, we see that $\gamma_k$ represents the best single scaling factor that relates the parameter update $\Delta \theta_k^t$ to the gradient $\nabla_k L(\theta^t)$ across the T time steps. In words – (i) $\nabla_k L(\theta^t)$ denotes the true direction of descent, (ii) $\Delta \theta_k^t$ denotes the actual direction as dictated by SGD. So, $\gamma_k$ captures how well the noisy gradient descent captures the true loss. If $\gamma_k \approx 0$, then the change in parameters is not related to the change in loss. And hence, it makes sense to prune these parameters.

**Second order analysis:**

For the second order analysis, we assume that we have reached a steady state distribution of the gradients $\nabla L(\theta)$ denoted by random variable **y** and $\Delta \theta$ denoted by random variable **x**. Hence we have that $E[\mathbf{x}] = 0$ i.e on an average the parameters do not change while there might be some variations at each step. Recall the second order taylor approximation,

$$\Delta L^t \approx \nabla L(\theta^{t-1})^T (\Delta \theta^t) + \frac{1}{2} (\Delta \theta^t)^T H(\theta^{t-1})(\Delta \theta^t) \tag{13}$$

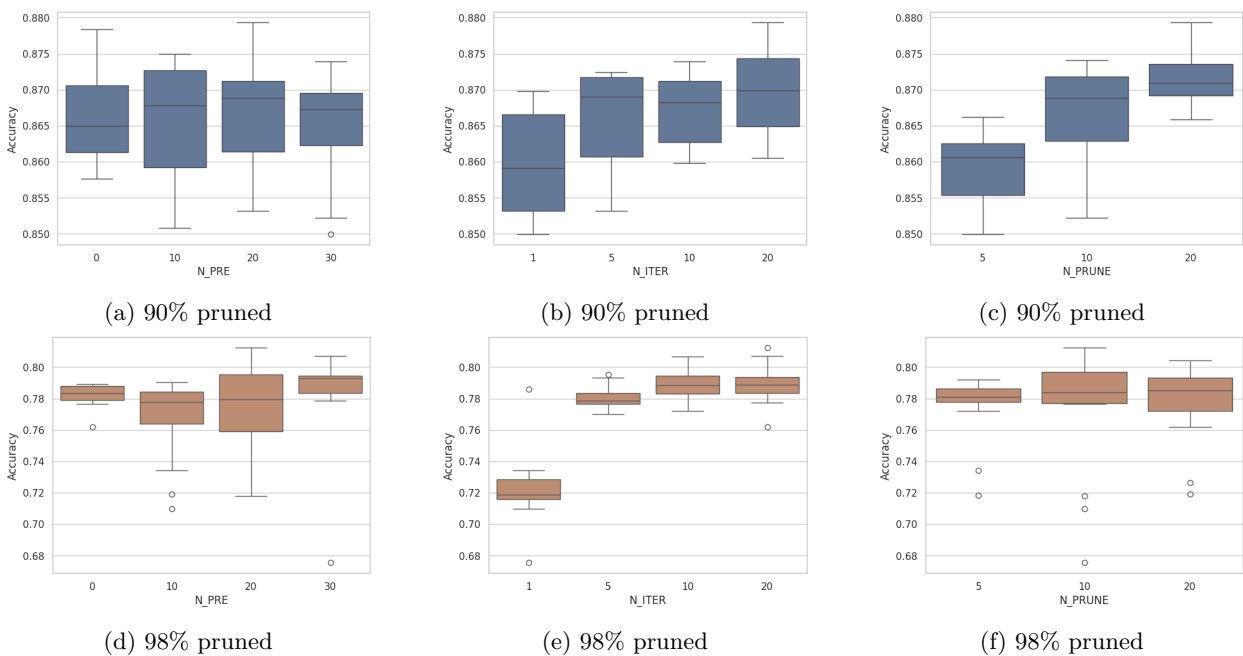

Figure 3: Effect of hyperparameters $N_{pre}$ $N_{iter}$, $N_{prune}$. These experiments have been performed using ResNet-20 on CIFAR10. Observe that accuracy remains robust to the changes in these hyperparameters. $N_{\text{pre}}$ determines how long the model is trained before pruning begins. The pruning schedule is controlled by $N_{\text{iter}}$, which sets how many rounds of pruning are applied. $N_{\text{prune}}$, defines how many epochs of training we use to record the SGD trajectory.

where $H(\theta^{t-1})$ denotes the hessian. We also assume for simplicity that the hessian does not change, i.e $H(\theta^t) \equiv H$. Again, to recall, causal pruning asks to fit the following (ignoring regularization),

$$\Delta L = \sum_k \gamma_k (\Delta \theta_k)^2 \tag{14}$$

Since, $E[\mathbf{x}] = 0$, we see that the following should hold true:

$$\gamma^* = \min_\gamma E[(\mathbf{x}^t H \mathbf{x} - \gamma^t \mathbf{x}^2)^2] = (E[\mathbf{x}^2 (\mathbf{x}^2)^t])^{-1} E[(\mathbf{x}^t H \mathbf{x}) \mathbf{x}^2] \tag{15}$$

**Key Insight:** Similar to the first order analysis, one can think of equation 15 as trying to explain the distribution $\mathbf{x}^t H \mathbf{x}$ using $\gamma^t \mathbf{x}^2$. So, after convergence the values of $\gamma$ tries to estimate the second order changes. If $\gamma_k = 0$ for some $k$, it implies that parameter dimension $k$ contributes negligibly to second-order changes (i.e., $\mathbf{x}^\top H \mathbf{x}$) and can therefore be pruned without significantly affecting the Hessian-based variation in the loss. Importantly, this implies that causal pruning does not change the flatness of the minima. We verify this in the experimental section.

## 5 Experiments

**Scope of Experiments:** Recall that the main aim of this article is to make explicit the implicit causal relation within gradient descent. We consider the application of pruning to explore the power of this observation. Our focus is not on achieving state-of-the-art results but on substantiating the claim above. The code for these experiments can be found at https://anonymous.4open.science/r/causalpruning-FA4A/README.md

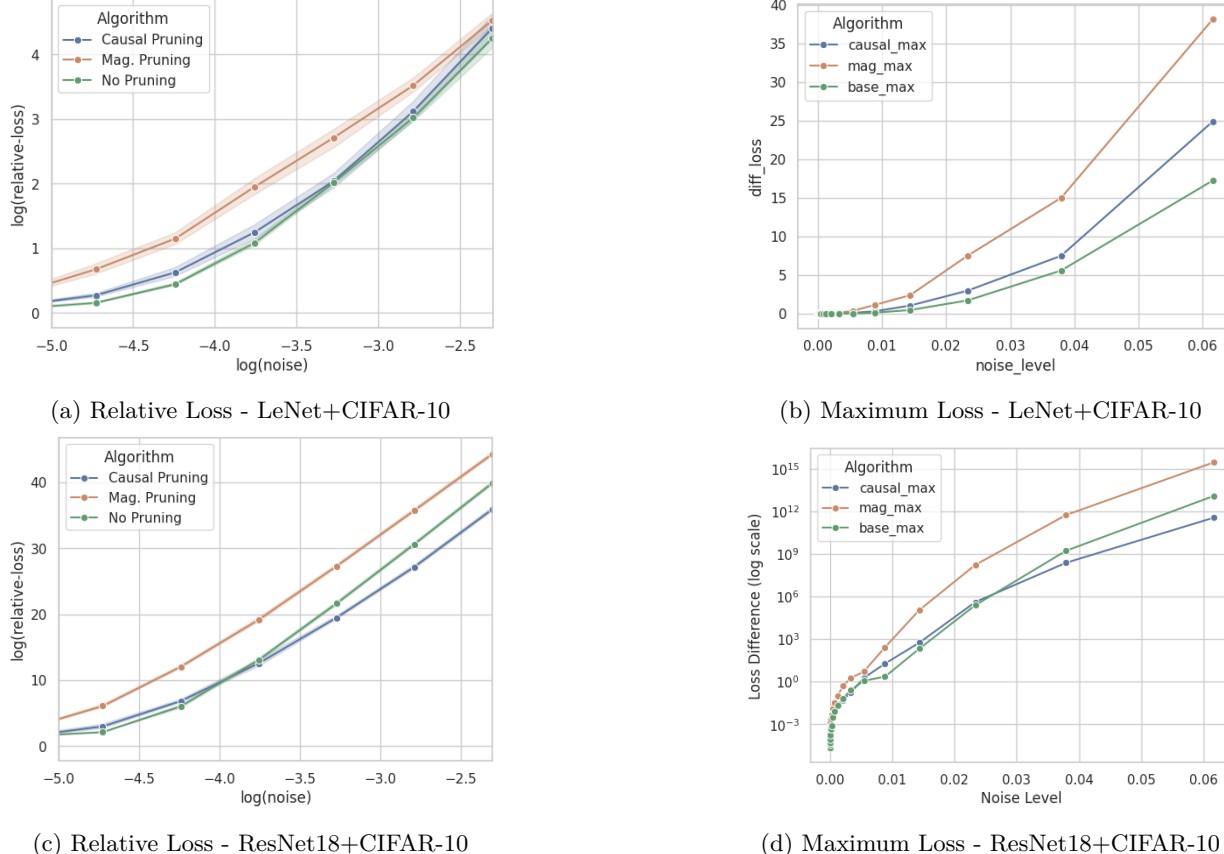

(a) Relative Loss - LeNet+CIFAR-10

(b) Maximum Loss - LeNet+CIFAR-10

(c) Relative Loss - ResNet18+CIFAR-10

(d) Maximum Loss - ResNet18+CIFAR-10

Figure 4: Causal pruning induces flatter minima compared to magnitude pruning and the unpruned baseline. Loss sensitivity is evaluated by perturbing trained parameters and measuring (a) the log-relative loss and (b) the maximum loss deviation over 100 perturbations. Results for LeNet (top row) and ResNet18 (bottom row) on CIFAR-10 show that causal pruning consistently leads to lower loss increases under perturbation, especially at larger noise levels. This indicates flatter local minima, whereas magnitude pruning results in significantly sharper minima.

## 5.1 Ablation on various parameters

To evaluate the sensitivity of Causal Pruning (CP) to its design choices, we varied three hyper-parameters $N_{\text{pre}}$, $N_{\text{iter}}$, and $N_{\text{prune}}$ —- and measured test accuracy at two sparsity levels: 90% and 98%. All experiments used ResNet-20 on CIFAR-10. After pruning we train the model to convergence to obtain the accuracies. Each configuration was run several times, and we summarize results using box plots in figure 3.

Each hyper-parameter controls a different stage of the CP pipeline. The parameter $N_{\text{pre}}$ determines how long the model is trained before pruning begins. This warm-up phase defines the basin of attraction where importance scores $\gamma_k$ are estimated. We explore values of $\{0, 10, 20, 30\}$. The pruning schedule is controlled by $N_{\text{iter}}$, which sets how many rounds of pruning are applied. After each round, we reset the remaining weights to the model state at epoch $N_{\text{pre}}$. Here, we consider $\{1, 5, 10, 20\}$. The third parameter, $N_{\text{prune}}$, defines how many epochs of training we use to record the SGD trajectory. This trajectory is then used to fit the Lasso model in equation 3, which estimates $\gamma_k$. We test $\{5, 10, 20\}$ for this parameter.

**Effect of $N_{\text{pre}}$:** At 90% sparsity (figure 3a), performance remains stable across the range of $N_{\text{pre}}$, with less than $\pm 0.3$ percentage points of variation. This suggests that CP does not require a lengthy warm-up to perform well. At 98% sparsity (figure 3d), a large value of $N_{\text{pre}} > 30$ seems to be preferable, even though it's not statistically significant.

**Effect of $N_{\mathbf{iter}}$:** Increasing the number of pruning iterations from 1 to 20 at 90% sparsity (figure 3b) leads to only a $\approx 1$ percentage point gain in accuracy. At 98% sparsity (figure 3e), the gains are more pronounced, with multi-step pruning yielding about $\approx 6$ percentage points of improvement. That said, the benefits saturate after roughly 10 iterations. Additional iterations help most in the high-sparsity regime but come with increased training time. For moderate sparsity levels like 90%, single-shot pruning often suffices. For extreme sparsity, we recommend using at least 10 rounds.

**Effect of $N_{\mathbf{prune}}$:** Varying the number of epochs used to record the SGD trajectory shows similar trends. At 90% sparsity (figure 3c), increasing $N_{\mathrm{prune}}$ from 5 to 20 epochs improves accuracy by about 1.5 percentage points. This indicates that the causal signal stabilizes quickly. At 98% sparsity (figure 3f), effect of longer windows is negligible. This shows that – small number of epochs are sufficient to identify the most important parameters, but one needs more "data" to identify parameters of lower importance precisely.

## 5.2 Causal Pruning Obtains a Flatter Minima

**Review of Flat Minima vs Generalization:** One popular hypothesis to explain the generalization of deep neural networks is the ideas of *flat minima*. The preference for flatter minima in deep learning often stems from their superior generalization performance, which can be understood by considering the inherent discrepancy between the training and test loss landscapes. Even if the test landscape is slightly shifted relative to the training one due to differences in data distribution, the model's parameters are likely to remain in a region of relatively low test loss (Neyshabur et al., 2017). Thus, flatness implies robustness to this train-test landscape shift, a key characteristic of models that generalize well (Jiang et al., 2020). Jiang et al. (2020) performs extensive experiments and shows that the *sharpness of the minima* is the most correlated measure with generalization. Foret et al. (2021) proposes a sharpness-aware minimization optimizer, which is shown to have better results than the rest of the minima. Ahn et al. (2024) uses the trace of the Hessian normalized by the dimension to measure the sharpness of the minima.

To assess the sharpness of minima induced by causal pruning, we evaluate on CIFAR-10 using two architectures of contrasting complexity – LeNet and ResNet18. This choice is motivated by the principle that demonstrating flatter minima at architectural extremes provides compelling evidence for the general flatness property.

We analyze the sensitivity of the loss function to parameter perturbations. Given a trained model with parameters $\theta$, we evaluate the perturbed loss $\mathcal{L}(\theta + \sigma\epsilon)$, where $\epsilon \sim \mathcal{N}(0, I)$ and $\sigma$ controls the perturbation scale. Two metrics are computed –

$$\text{Relative Loss} = \log\left(\frac{\mathcal{L}(\theta + \sigma\epsilon)}{\mathcal{L}(\theta)}\right) \tag{16}$$

averaged across perturbations, measuring average curvature around the minimum. And,

$$\text{Maximum Loss} = \max_j |\mathcal{L}(\theta + \sigma\epsilon_j) - \mathcal{L}(\theta)| \tag{17}$$

over 100 noise realizations, quantifying worst-case sensitivity. Figure 4 present these metrics for LeNet and ResNet18 trained on CIFAR-10, comparing causal pruning, magnitude pruning, and the unpruned baseline.

Across both architectures, causal pruning consistently yields lower relative and maximum loss deviations than magnitude pruning, especially at higher noise levels. This behavior is particularly pronounced in the ResNet18 results, where causal pruning maintains exponentially smaller maximum losses compared to magnitude pruning as well as unpruned baselines (figure 4d).

These trends provide empirical support to the statement – *causal pruning leads to flatter minima*. In sharp minima, even small perturbations in parameter space lead to steep increases in loss, whereas flatter minima exhibit broader, low-loss basins. The consistent suppression of both average and worst-case loss increases under perturbation for causal pruning indicates a superior alignment with flatter regions of the loss landscape.

## 5.3 Causal vs Magnitude Heuristic

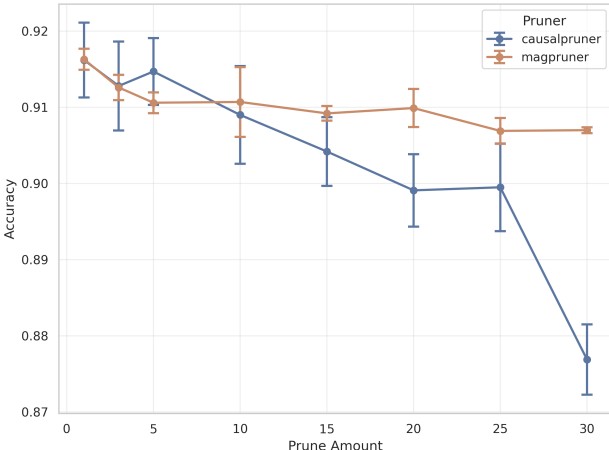

Figure 5: Comparison of causal pruning vs. magnitude-based pruning on ResNet-20/CIFAR-10. We evaluate both methods by removing the highest-scoring parameters (inverse pruning) to assess heuristic quality. Superior heuristics should exhibit more dramatic performance degradation when their most important parameters are removed.

We compare causal pruning against magnitude-based pruning heuristics. Both methods assign importance scores to network parameters – causal coefficients in the former case and absolute parameter magnitudes in the latter – and remove parameters with the lowest scores. To evaluate the relative effectiveness of these heuristics, we perform a stress test by removing parameters with the *highest* scores instead. Under this inverse pruning regime, the superior heuristic should exhibit more dramatic performance degradation, as removing its most important parameters should be more detrimental to model performance.

**Experimental Setup** We evaluate our approach using ResNet-20 on CIFAR-10. We exclude the final fully connected layer from pruning as it comprises only 640 parameters (0.03% of total parameters) yet represents the most critical components for classification. Removing these parameters would result in catastrophic performance collapse, obscuring meaningful comparisons between pruning heuristics. We systematically vary the pruning ratio from 1% to 30% across the remaining convolutional layers. To isolate the fundamental differences between pruning heuristics, we restrict our evaluation to single-shot pruning – iterative approaches introduce additional hyperparameters and optimization dynamics that would confound the core comparison between causal and magnitude-based heuristics.

Figure 5 validates our hypothesis through inverse pruning evaluation. While both methods perform similarly at low pruning ratios (1-10%), causal pruning exhibits significantly steeper performance degradation beyond 10% – dropping 3.2 percentage points (0.914 to 0.882) compared to magnitude pruning's 0.9 percentage point decline (0.916 to 0.907). This counter-intuitive result confirms causal pruning's superiority: the dramatic collapse when removing high-importance parameters indicates more accurate identification of truly critical weights, whereas magnitude pruning's gentler decline suggests it conflates parameter scale with functional importance.

### 5.4 Comparison with other baselines

**Baselines:** To the best of our knowledge, there does not exist any methods which explore the role of gradient trajectories in pruning. Nevertheless, for completeness we compare causal pruning against a diverse suite of state-of-the-art pruning methods – including CHITA, CHITA++ (Benbaki et al., 2023), CBS (Yu et al., 2022), CNN-CFC(Li et al., 2019), NN-Relief(Dekhovich et al., 2024), DRIVE(Saikumar & Varghese, 2024), SNIP(Lee et al., 2019), and HRank(Lin et al., 2020) – across multiple architectures and datasets under both single-shot and iterative pruning paradigms. The results consistently demonstrate the robustness and

Table 1: Comparing Causal Pruning with Baselines (Single Shot). ▮ indicates best, ▮ indicates second-best.

| Model (# Params) Dataset | % Pruned | Algorithm | Accuracy | | |
|---|---|---|---|---|---|
| | | | Baseline | Pruned | $\Delta$ |
| ResNet-20 (200k) CIFAR10 | 80 | CHITA | 91.36 | 57.90 | 33.46 |
| | | CHITA++ | 91.36 | 88.72 | 2.64 |
| | | CBS | 91.36 | 51.28 | 40.08 |
| | | Causal Pruning ($N_{\text{iter}} = 1$) | 91.42 | 86.54 | 4.88 |
| | 90 | CHITA | 91.36 | 15.6 | 75.76 |
| | | CHITA++ | 91.36 | 79.32 | 12.04 |
| | | Causal Pruning ($N_{\text{iter}} = 1$) | 91.42 | 82.49 | 8.93 |
| MobilenetV1 (3M) Imagenet | 80 | CHITA | 71.95 | 29.78 | 42.37 |
| | | CHITA++ | 71.95 | 47.45 | 24.50 |
| | | CBS | 71.95 | 16.38 | 55.57 |
| | | Causal Pruning ($N_{\text{iter}} = 1$) | 71.95 | 56.82 | 15.13 |
| ResNet50 (25M) Imagenet | 80 | CHITA | 77.01 | 45.00[2] | 32.01 |
| | | Causal Pruning ($N_{\text{iter}} = 1$) | 77.01 | 67.95 | 9.06 |

Table 2: Comparing Causal Pruning with Baselines (Iterative). ▮ indicates best, ▮ indicates second-best.

| Model (# Params) Dataset | % Pruned | Algorithm | Accuracy | | |
|---|---|---|---|---|---|
| | | | Baseline | Pruned | $\Delta$ |
| ResNet-20 (200k) CIFAR10 | 43 | CNN-CFC | 92.20 | 91.13 | 1.07 |
| | 64 | NN-Relief | 92.25 | 91.10 | 1.15 |
| | 70 | Magnitude Pruning (IMP) | 92.4 | 91.2 | 1.2 |
| | 70 | Causal Pruning | 91.42 | 90.52 | 0.90 |
| VGG (16M) CIFAR10 | 82 | HRank | 93.96 | 92.34 | 1.62 |
| | 95 | DRIVE | 92.40 | 92.68 | -0.28 |
| | 95 | Magnitude Pruning | 92.40 | 90.20 | 2.2 |
| | 95 | Causal Pruning | 93.30 | 92.90 | 0.40 |
| | 98 | NN-Relief | 92.50 | 92.40 | 0.10 |
| | 97 | SNIP | 91.70 | 92.00 | -0.30 |
| | 98 | DRIVE | 92.40 | 91.36 | 1.04 |
| | 98 | Magnitude Pruning | 92.40 | 90.04 | 2.36 |
| | 98 | Causal Pruning | 93.30 | 92.36 | 0.60 |
| VGG (16M) Tiny-Imagenet | 95 | SNIP | 45.14 | 44.27 | 0.87 |
| | 95 | DRIVE | 48.74 | 45.60 | 3.14 |
| | 95 | Magnitude Pruning | 48.74 | 46.79 | 1.95 |
| | 95 | Causal Pruning | 56.01 | 54.31 | 1.70 |
| | 97 | NN-Relief | 45.63 | 45.60 | 0.03 |
| | 98 | DRIVE | 48.74 | 42.25 | 6.49 |
| | 98 | Magnitude Pruning | 48.74 | 44.26 | 4.48 |
| | 98 | Causal Pruning | 56.01 | 50.04 | 5.97 |

efficacy of causal pruning, particularly in high-sparsity regimes and on challenging datasets such as Tiny-ImageNet.

**Remark:** We evaluate causal pruning in both single-shot and iterative settings against diverse baselines. Since causal pruning introduces a novel principle – leveraging observational gradient dynamics for parameter

importance – we compare against established methods where they perform optimally. We categorize baselines as single-shot or iterative based on their optimal performance setting. Further, results are taken directly from respective papers, matching the nearest pruning ratios and baseline accuracies.

**Single Shot Performance:** In the single-shot setting (Table 1), causal pruning outperforms or closely trails CHITA++ - the strongest baseline in preserving post-pruning accuracy. For ResNet-20 on CIFAR-10 at 80% sparsity, CHITA++ achieves the smallest accuracy drop ($\Delta = 2.64$), with causal pruning a close second ($\Delta = 4.88$), and significantly outperforming CHITA ($\Delta = 33.46$) and CBS ($\Delta = 40.08$). However, at 90% sparsity, causal pruning ($\Delta = 8.93$) surpasses CHITA++ ($\Delta = 12.04$), which shows its ability to maintain performance under aggressive compression.

A similar pattern is observed on MobileNetV1 for ImageNet at 80% sparsity: causal pruning retains 56.82% top-1 accuracy, a substantial improvement over CHITA (29.78%), CBS (16.38%), and even CHITA++ (47.45%). This represents a $\Delta$ of only 15.13 for causal pruning. Notably, on ResNet50 (ImageNet, 80% pruning), causal pruning exhibits a $\Delta$ of 9.06 compared to CHITA's 32.01, demonstrating a pronounced advantage in deeper architectures.

Across all single-shot experiments, causal pruning shows a consistent trend of superior accuracy retention at high pruning ratios, especially on larger networks and complex datasets.

**Iterative Pruning Performance:** In the iterative pruning regime (Table 2), causal pruning remains competitive, with performance gaps narrowing. This is due to the fact that, all the approaches work more-or-less similarly at low levels of pruning. For ResNet-20 on CIFAR-10, causal pruning achieves the lowest $\Delta$ (0.90), outperforming CNN-CFC (1.07) and NN-Relief (1.15), despite operating at a higher pruning ratio (70%).

On VGG with CIFAR-10, causal pruning trails only marginally behind DRIVE and SNIP at extreme pruning levels. At 95% sparsity, causal pruning attains $\Delta = 0.40$ versus DRIVE's $\Delta = -0.28$ (where a negative $\Delta$ indicates improved accuracy post-pruning). At 98% pruning, causal pruning's $\Delta = 0.60$ is second-best behind SNIP ($\Delta = -0.30$), still outperforming DRIVE ($\Delta = 1.04$). These results highlight causal pruning's robustness even in highly iterative settings, where certain baselines benefit from fine-grained gradient and Hessian information.

On the more challenging Tiny-ImageNet dataset with VGG, causal pruning markedly surpasses other methods. At 95% sparsity, it retains 54.31% accuracy ($\Delta = 1.70$), well above DRIVE ($\Delta = 3.14$). At 98% pruning, causal pruning ($\Delta = 5.97$) again significantly improves upon DRIVE ($\Delta = 6.49$) suggesting better generalization.

Overall, causal pruning consistently matches or outperforms the some of the best-performing baselines across architectures, datasets, and pruning regimes. Although the results are mixed when compared to state-of-the-art.

## 6   Conclusion

This explicit causal model equation 3 provides a natural framework for network pruning. Existing network pruning criteria involve either assessing the sensitivity of the loss function to a weight's removal (LeCun et al., 1989; Singh & Alistarh, 2020) ) or evaluating the magnitude of the weight itself (Wang et al., 2023; Han et al., 2016). In contrast, we utilize observational values to decide usefulness. The core principle is simple – if a parameter $\theta_k$'s changes $(\Delta\theta_k)^2$ do not consistently contribute to the observed loss reduction $\Delta L$, its learned causal coefficient $\gamma_k$ should be close to zero. Such parameters are effectively "causally unimportant" within the recent training dynamics and can be pruned.

The theoretical analysis reveals two primary insights. First, we derive causal coefficients $\gamma_k$, which quantify how changes in parameters affect the loss function. At convergence, these coefficients provide an optimal diagonal approximation to the Hessian, thus preserving the flatness of the minima. We empirically validate this flatness property using two neural network models subjected to Gaussian perturbations.

Empirically, causal pruning achieves performance competitive with state-of-the-art methods, significantly outperforming existing approaches in single-shot pruning scenarios, though this advantage diminishes under iterative pruning conditions. Additionally, we extend our framework to optimizers like SGD with Momentum, introducing more complex linear models involving terms such as $(\Delta\theta_k^t)^2$ and cross-terms $(\Delta\theta_k^t)(\Delta\theta_k^{t+1})$. This extended model remains solvable using Lasso regression, where parameter-specific causal coefficients $\gamma_k$ are learned to identify and prune unimportant parameters effectively.

To the best of our knowledge, this approach provides a novel interpretation of gradient descent dynamics through the lens of explicit, learnable Granger-type causality. *In the process, we also achieve pruning from the lens of causality via sparsity (induced by L1 regularization)*! Future work will explore further extensions of this causal pruning framework to adaptive optimization methods such as Adam (Kingma & Ba, 2015).

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

## A   Deriving the Causal Relation in the case of GD + Momentum

Let's consider the following variant of the gradient descent with momentum-

$$v^{t+1} = \beta v^t + \frac{\partial L(\theta^t)}{\partial \theta}$$
$$\theta^{t+1} = \theta^t - \eta v^{t+1} \tag{18}$$

where $\beta$ denotes the momentum hyperparameter. Then, we have, using first-order Taylor approximation,

$$L(\theta^{t+1}) = L(\theta^t) - \eta \left(\frac{\partial L(\theta^t)}{\partial \theta}\right)^T v^{t+1}$$
$$= L(\theta^t) + \eta\beta \left(v^{t+1}\right)^T v^t - \eta \left(v^{t+1}\right)^T v^{t+1} \tag{19}$$

We also have,

$$v^{t+1} = \frac{\theta^{t+1} - \theta^t}{\eta} = \frac{\Delta\theta^t}{\eta} \quad \Rightarrow \eta \left(v^{t+1}\right)^T v^{t+1} = \frac{\|\theta^{t+1} - \theta^t\|^2}{\eta} \tag{20}$$

$$v^{t+1} = \eta v^t + \frac{\partial L(\theta^t)}{\partial \theta} \Longrightarrow \langle v^{t+1}, v^t \rangle = \eta\|v_t\|^2 + \left\langle \frac{\partial L(\theta^t)}{\partial \theta}, v_t \right\rangle \tag{21}$$

$$\left\langle \frac{\partial L(\theta^t)}{\partial \theta}, v_t \right\rangle = \frac{1}{\eta} \left\langle \frac{\partial L(\theta^t)}{\partial \theta}, (\theta^{t-1} - \theta^t) \right\rangle \approx L(\theta^{t-1}) - L(\theta^t) \tag{22}$$

Substituting, equations 20,21,22 in equation 19, we get,

$$L(\theta^{t+1}) = L(\theta^t) - \frac{1}{\eta} \sum_k (\theta_k^{t+1} - \theta_k^t)^2 + \eta^2 \sum_k (\theta_k^t - \theta_k^{t-1})^2 - \eta\beta(L(\theta^{t-1}) - L(\theta^t)) \tag{23}$$

Ignoring the constants $\beta, \eta$, and replacing them with $c_1, c_2, c_3.c_4$,

$$L(\theta^{t+1}) = c_1 L(\theta^t) + c_2 L(\theta^{t-1}) + c_3 \sum_k (\theta_k^{t+1} - \theta_k^t)^2 + c_4 \sum_k (\theta_k^t - \theta_k^{t-1})^2 \tag{24}$$

where $c_1, c_2, c_3, c_4$ are some fixed functions of $\eta, \beta$. Using the same principle above, and replacing the coefficients with parameter specific $\gamma_k$. However, note that here we have two possibly independent features for each parameter $\theta_k$ - one for each time difference. Hence instead of using a single parameter $\gamma_k$ (as in the case of vanilla gradient descent), one needs to use $(\gamma_{k,0}, \gamma_{k,1})$. The model then becomes,

$$L(\theta^{t+1}) = c_1 L(\theta^t) + c_2 L(\theta^{t-1}) + \sum_k \gamma_{k,0}(\theta_k^{t+1} - \theta_k^t)^2 + \sum_k \gamma_{k,1}(\theta_k^t - \theta_k^{t-1})^2 \qquad (25)$$

Here we consider the parameter to be not-important if $\gamma_{k,0} = \gamma_{k,1} = 0$, i.e all the coefficients should be irrelevant. We also replace the causal pruning step accordingly.

**Remark: What about the case with varying learning rates?** It turns out that it is not easy to adapt the above procedure to gradient descent with adaptive learning rates such as ADAM (Kingma & Ba, 2015) or RMSProp (Hinton et al., 2014). Specifically, one would have to consider all the updates till time $t$ - $\{\Delta\theta^t\}_{t=0}^{t=t}$. Since this is computationally expensive and also since we see that GD with momentum works sufficiently well in practice (Loshchilov & Hutter, 2017), we do not consider this case in the current scope of the article.

## B  Details of Flat Minima Experiments

To obtain the top eigenvalues from trained networks, we use the stochastic power-iteration method from Yao et al. (2020). For visualizing the minima, we use the method proposed in Li et al. (2018), which is the following:

1. We consider two arbitrary directions by initializing a network with weights from a normal distribution with mean 0 and standard deviation 1 - $u_1$ and $u_2$. We then normalize the filters to have the same norm as the original network

$$u_{i,j} = \frac{u_{i,j}}{\|u_{i,j}\|}\|d_{i,j}\| \qquad (26)$$

where $u_{i,j}$ refers to the $j^{th}$ filter from $i^{th}$ layer of randomly initialized network, and $d_{i,j}$ refers to the the $j^{th}$ filter from $i^{th}$ layer of the trained network.

2. We then plot the loss obtained by $f(\theta^* + \alpha u_1 + \beta u_2)$ where $f$ refers to the network architecture, $\theta^*$ refers to the parameters of the original network and $\alpha, \beta \in (-1, 1)$. This is visualized as a 3d plot as shown in the figures. Further, we also scale the z-axis as $\log(1 + f(\theta))$ for better visualization.

## C  Experimental Details

All experiments were carried out on a NVIDIA H100 server. The server is equipped with 2 AMD EPYC 9454 48-Core Processor with 48 physical cores per socket totaling 192 threads, paired with 512 GB of system memory. The server houses a NVIDIA H100 NVL GPU with 94GB of VRAM.

**Hyperparameter Settings**

This section outlines the hyperparameter configurations used across all experiments evaluating causal pruning. Our experimental scope spans multiple architectures and datasets, including CIFAR-10, Tiny ImageNet, and ImageNet, with both randomly initialized and pretrained models.

**Optimizer and Learning Rate Scheduling:** Except for experiments on ImageNet, we use SGD with momentum as the default optimizer. The learning rate schedule is either cosine annealing or one-cycle LR. We set the base learning rate to 0.001, and the maximum learning rate to 0.1. All models are trained with a fixed momentum of 0.9.

| Hyperparameter | Symbol | Range/Values |
|---|---|---|
| Warm-up epochs before pruning | $N_{\mathrm{pre}}$ | 0, 1, 10, 20, 30, 60 |
| Pruning iterations | $N_{\mathrm{iter}}$ | 1, 5, 10, 20, 30, 40 |
| Epochs to collect SGD trajectory | $N_{\mathrm{prune}}$ | 1, 5, 10, 20 |
| Total pruning ratio | $P$ | 0.1 to 0.993 |
| Causal Lasso max iterations | — | 50 |
| Tolerance for convergence in Lasso | — | 5 (no improvement steps) |

Table 3: Range of hyperparameters used across causal pruning experiments.

**Optimization for Lasso Regression in equation 3:**  As described in the main text, we solve the lasso regression using the stochastic method from Shalev-Shwartz & Tewari (2011). We use a learning rate of 0.1, with a maximum of 50 iterations and a convergence tolerance of 5 iterations. The batch size is fixed to 16 due to memory constraints.

All reported accuracies correspond to pruned models that are trained to convergence. We report the best accuracy achieved during this phase.

**Configuration for VGG-Style Architectures:**  On CIFAR-10, we compare an unpruned baseline against models pruned to 98% sparsity using causal pruning over 30 iterations. On Tiny ImageNet, we adopt a more extended schedule: pruning begins after 60 warm-up epochs, followed by a 120-epoch pre-pruning phase. We collect SGD trajectories over 10 epochs per iteration, across 30 pruning iterations, targeting 98% sparsity.

For VGG-style models pretrained on CIFAR-10, we use a shortened pre-prune phase of 10 epochs and initialize the causal-specific learning rate to $10^{-4}$.

**Large-Scale Architectures and Datasets:**  On ImageNet, we apply causal pruning to ResNet50 and MobileNetV1, both pretrained. We prune to 80% sparsity in a single iteration, using only one warm-up and one pre-prune epoch. This minimal setup highlights the practicality of rapid causal pruning at scale. Post-pruning training is performed with AdamW, using a learning rate of 0.01 and a weight decay of 0.01. We fine-tune each model for at most 30 epochs.

