# OpenReview forum: "Beyond Magnitude and Gradient: Network Pruning Inspired by Optimization Trajectories"
_TMLR — Rejected by TMLR_

### Review · Reviewer_GChL · 2025-08-11

**Summary Of Contributions:**

This paper introduces Causal Pruning (CP), a novel method for network pruning that learns parameter importance from the optimization trajectory. The core idea is to model the relationship between parameter changes and the resulting loss change during training. By fitting a sparse linear model using Lasso regression, the method identifies parameters with negligible "causal" impact on loss reduction and prunes them.

**Audience:**

Yes

**Audience Explanation:**

The core idea of learning parameter importance from optimization trajectories is novel and conceptually interesting. Researchers in model compression, optimization, and deep learning theory would be intrigued by this different perspective on pruning. The connection to flat minima is also a compelling angle.

However, the TMLR audience would also expect rigorous, modern, and convincing empirical validation. While the idea itself is valuable, its current presentation and lack of robust evidence would leave readers questioning its practical utility and true standing relative to other methods. Therefore, while the initial concept would attract interest, the paper in its current form would not satisfy the community's standards for empirical rigor.

**Broader Impact Concerns:**

I do not have any broader impact concerns regarding this work.

**Claims And Evidence:**

No

**Claims Explanation:**

While the paper presents a novel idea with some supporting evidence, several significant issues with the experimental validation prevent the claims from being fully convincing.

1.  **Inconsistent and Non-Standard Experimental Setups:** The evidence used to support the method's superiority is based on questionable experimental choices.
    *   Figure 2 uses LeNet trained on CIFAR-10. LeNet was designed for the much simpler MNIST dataset and is not a suitable or standard backbone for CIFAR-10. Conclusions drawn from this setup are not representative of modern deep learning practices.
    *   Figure 4 uses ResNet-18 on CIFAR-10 to demonstrate flatness. However, the original ResNet paper introduced specific ResNet architectures for CIFAR-10 (e.g., ResNet-20, ResNet-32). Table 1 correctly uses ResNet-20 for its CIFAR-10 benchmark. This inconsistency across the paper's own experiments makes the results difficult to interpret and compare. A proper scientific comparison requires a consistent and standard experimental backbone.

2.  **Incomplete Comparison to State-of-the-Art:** The claim of outperforming CHITA on ResNet-50 for ImageNet (Table 1) is based on a single pruning ratio (80%). The CHITA paper (Figure 2a) shows performance across a wide range of sparsities. A single data point is insufficient to claim superiority and could be the result of cherry-picking a favorable setting. Furthermore, the footnote attached to the CHITA accuracy (45.002) is not explained in the paper, making the result irreproducible and unclear.

3.  **Lack of Evidence for Modern Architectures:** The paper makes general claims about pruning neural networks but provides no evidence for its effectiveness on Transformer models. In the current research landscape, a pruning method that has not been validated on Transformers cannot be considered state-of-the-art or broadly applicable. This is a major omission that significantly weakens the paper's overall claims.

In summary, the core idea is somewhat interesting, but the evidence provided is weakened by experimental design, incomplete comparisons, and a narrow architectural scope.

**Requested Changes:**

To secure a recommendation for acceptance, the following changes are critical. They address fundamental flaws in the experimental validation and scope of the paper.

**Critical Changes:**

1.  **Standardize and Justify Experimental Setups:** The experimental design must be made consistent and adhere to community standards.
  - Replace the LeNet-on-CIFAR10 experiment (Figure 2). If the goal is to show the Hessian spectrum, this should be done on a standard and consistent setup used elsewhere in the paper, such as ResNet-20 on CIFAR-10.
 - Unify the ResNet architecture for CIFAR-10 experiments. All CIFAR-10 experiments (including the flatness analysis in Figure 4) should use the same backbone (e.g., ResNet-20, as in Table 1) for consistency and comparability.

2.  **Provide a Comprehensive and Fair Comparison to SOTA:** The comparison with CHITA on ImageNet must be more thorough.
   - Expand the ResNet-50 on ImageNet comparison in Table 1. Instead of a single point at 80% sparsity, provide results across a range of sparsities to allow for a fair comparison against the performance curve shown in the CHITA paper.
    -Clarify the footnote for the CHITA accuracy (45.002). All numbers from other papers must be clearly sourced and explained.

3.  **Demonstrate Applicability to Transformer Architectures:** To claim broad relevance in 2024/2025, the method's effectiveness must be demonstrated on modern architectures.
  - Add at least one set of experiments on a standard Transformer-based model. For example, pruning a Vision Transformer (ViT) on ImageNet or a BERT-family model on a GLUE benchmark task would be sufficient to demonstrate the method's potential beyond CNNs.

**Strengthening Changes (Not critical, but would improve the paper):**

1.  **Quantify and Discuss Practical Overhead:** The paper should be more transparent about the practical costs of the method.

- Add a quantitative discussion of the storage and computational overhead. For a large model like ResNet-50, specify the amount of temporary storage needed to save the trajectories and the additional time required for the Lasso regression step. This provides crucial context for practitioners.

---

> ### Author Response · Authors · 2025-08-16
>
> We thank the reviewer for their detailed feedback and the recognition that our core idea is "novel and conceptually interesting."
>
> Before addressing specific concerns, we would like to re-iterate our claims and primary contribution, which provides context for our experimental choices.
>
> **Main Claim of the article:** The lasso regression coefficients ($\gamma_k$) from the SGD optimization trajectories have ``causal'' information regarding the importance. We find this observation *surprising*, since there is no a priori reason that these quantities ($\gamma_k$ and importances) would align. The proof-of-concept for this is given in figure 1.
>
> This insight gives us a fundamentally new perspective on understanding parameter importance through optimization dynamics. Our experiments demonstrate that this connection translates into a viable pruning method that performs competitively with existing approaches, thereby validating our observations.
>
> We would also like to re-emphasize that -- we neither claim nor aim that our approach is state-of-the-art (Page 8, *scope of experiments* paragraph). However, as stated in the TMLR acceptance criteria:
>
> ```
> Crucially, it should not be used as a reason to reject work that isn't considered ‘significant’ or ‘impactful’ because it isn't achieving a new state-of-the-art on some benchmark. Nor should it form the basis for rejecting work on a method considered not ‘novel enough’, as novelty of the studied method is not a necessary criteria for acceptance.
> ```
>
> Our work aims to make technically sound and interesting contributions, and we believe it aligns with this criterion.
>
> **1. Experimental Setup in Section 5.2 vs Section 5.3**
>
> The aim of section 5.2 is to show that we obtain a flatter minima. To show this empirically, we
>
> - Consider a common dataset -- CIFAR10
> - Two architectures of extreme sizes -- LeNet and Resnet18 -- First one with only small number of layers and the second which is vastly over-parametrized.
>
> This motivated our the description in section 5.2. Further, we also respectfully disagree that -- Since LeNet is designed for MNIST, we must not use it for CIFAR10. The whole point of an architecture is its ability to be tested on diverse data sets.
>
> Since the aim of section 5.3 is to validate our approach with existing pruning methods (very different objective than section 5.2), we considered the standard settings as the reviewer suggested.
>
> In our opinion, we believe that the above experimental design is sound. We shall clarify this in the revised version.
>
> **2. CHITA Comparison and Scope**
>
> We apologize for the missing information. The missing footnote was intended to indicate that the CHITA result was "deduced from Figure 2a of the original paper." We shall fix this in the revision.
>
> Unfortunately, we cannot run the CHITA method with our resource constraints. The original article only provides results up to 80\% pruning. We could include results at lower pruning percentages (70\%), but this comparison would not add any new information: CHITA reports (approx) 66\% accuracy at 70\% pruning, while we achieve 67\% accuracy at the more aggressive 80\% pruning level. Given that this would demonstrate similar competitive performance without adding substantial new insights, we focused our limited space on the current comparison.
>
>
> **3. Modern Architecture Limitations**
>
> To recall, we do not aim for state-of-the-art in this article. Moreover, given the cost of running large-scale experiments (ViT on ImageNet-21K) and the fact that CNNs remain the appropriate architecture choice for the datasets and scales we investigate, we consider Transformer validation outside the scope of this article.
>
> Rather than claiming universal applicability, we demonstrate the fundamental principle of causal pruning. The theoretical insight itself is architecture-agnostic, though full validation across all modern architectures remains future work.
>
> **4. Computational Considerations**
>
> We have provided the analysis of computational overhead on page 5 (paragraph **Computational Complexity**), addressing the practical implementation costs of our approach. Please let us know if further information is required, and we shall add it in the revised version.
>
>
> In conclusion, our work demonstrates a theoretically surprising and practically viable connection between optimization trajectories and parameter importance. The experimental validation confirms that this insight leads to competitive pruning performance while opening new avenues for understanding the relationship between optimization dynamics and network compression.

---

### Review · Reviewer_Hx2Z · 2025-08-14

**Summary Of Contributions:**

This paper proposes a novel criterion (a metric used to decide the importance of parameters in a neural network) by watching how much they contribute to the loss during training rather than at the magnitude or gradient. It uses lasso regression to identify which parameters have strong vs weak casual relationships with respect to loss reduction during training, and prunes parameters with smallest yk coefficient. It opens up a new direction for understanding parameter importance for pruning.

**Audience:**

Yes

**Audience Explanation:**

I can see this work being helpful to the community should the limitations be resolved. Pruning criteria are widely applicable across any pruning and sparse training pipelines, should it be proven effective with more experiments. The connection between Granger causality and training dynamics (albeit flawed) couldbe compelling to some theory-driven researchers.

**Claims And Evidence:**

No

**Claims Explanation:**

Granger Causality issues:

- The paper's Granger causality approach is problematic. Granger causality assumes stationarity, meaning the statistical properties of the time-series don't change over time, ie., mean, variance, relationships don't shift over time. Network training is dynamic with distinct phases, ie., faster convergence in the beginning. It violates the stationary time-series assumption required for Granger causality.

- Section 4.1 "From Causality to Pruning via Sparsity ..." - This is a logical leap. Granger causality describes a predictive relationship, but pruning is about functional importance. A parameter could have low yk for many reasons unrelated to functional importance: noisy gradients, correlated updates with other parameters, being in a different learning phase, or having small but critical contributions. Conversely, a parameter could have high yk but be functionally redundant due to network over-parameterization. I would be more convinced if the paper demonstrated empirical evidence, that pruning low yk gives better performance than pruning high yk. This is a fundamental key experiment that is missing. The closest thing it has is Figure 1, which shows yk coefficients with lasso importance on Friedman dataset, but this is not a neural network, not pruning.


Evaluation issues:

- Performance comparison is somewhat not apple-to-apple. For instance, why aren't the "% pruned" consistent for all methods in Table 2? (ResNet-20: CHITA at 70% vs CNN-CFC at 43% vs NN-Relief at 64%) - this is not very helpful for telling which is better.

- The paper's title is "beyond magnitude", but it seems the only place it actually compares with it is Figure 2b, at 90% sparsities on Cifar10 only. What about other configurations? Unless I missed it, one sparsity, one architecture, one dataset isn't enough for such a crucial benchmark.

- Multiple sparsities should be used for one-shot in Table 1 (ie., {70%,80%,90%})

- SNIP is a one-shot method, why is it in Table 2?

- Hrank is structured pruning, but CP is unstructured, so this isn't fair!

- It also lacks confidence intervals and multi-seed evaluation.

**Requested Changes:**

- Add ablation, showing pruning low yk > high yk. ie., sort the yk list, show remove top X% vs remove bottom X%.

- Magnitude which is still the king in terms of performance vs complexity trade off and should be more rigorously compared against. I would be more convinced if CP beats magnitude on either one-shot or iterative, at different sparsities {90%, 95%, 98%}. It would be helpful to include random pruning as a reference point.

- Consider including Cifar100 and/or object detection.

- The method requires collecting SGD trajectories and solving Lasso regression - it could use some analysis of the additional computational/memory costs compared to simpler baselines.

- Provide rigorous justification for why Granger causality coefficients reflect functional importance, or reframe the method without causal claims.

---

> ### Author Response · Authors · 2025-08-16
>
> We thank the reviewer for their detailed feedback and recognition that our work "opens up a new direction for understanding parameter pruning...".
>
> **Core Contribution and Granger Causality Framework**
>
> Before addressing specific points, we clarify our use of Granger causality. We employ this framework as an intuitive explanation for why SGD trajectory data relates to parameter importance, rather than claiming formal statistical causality. To recall, our $\gamma_k$ coefficients are derived from the model $\Delta L = \sum_k \gamma_k (\Delta \theta_k)^2$ fitted to observed SGD dynamics.
>
> To make the stationary assumption reasonable we use **short training runs** and **resetting parameter weights**. Although this is not perfect, empirical results show it does a reasonable job of obtaining good estimates.
>
> **1.Experiment: High vs Low** $\gamma_k$ **Pruning**
>
> We thank the reviewer for this simple but effective idea, which tests our claim better. We did the experiments on resnet20 + CIFAR10 and observed that *pruning high* $\gamma_k$ *parameters caused catastrophic model collapse*.
>
> Pruning just 30\% of the high $\gamma_k$ values reduced the accuracy to 10\%. While pruning 30\% of low $\gamma_k$ values maintains the same performance as the original network. This further supports the claim that $\gamma_k$ coefficients capture functional importance.
>
> We would like to thank the reviewer (once again) for this experiment, and we shall include this in the revised version.
>
> **2. Granger Causality and Non-Stationarity Concerns**
>
> We acknowledge that neural network training violates strict stationarity assumptions. However, our approach mitigates this by collecting data over short periods (single epoch) where local stationarity is more reasonable. Moreover, the empirical validation (Sec 5.3) demonstrates that despite theoretical limitations, the method does captures meaningful relationships. The model collapse experiment further adds to evidence that $\gamma_k$ reflects the functional importance.
>
> **3. Magnitude Pruning Comparisons**
>
> We would like to emphasize that, we do not aim for state-of-the-art in this article. Our method achieves competitive performance while offering theoretical insights beyond magnitude-based approaches.
>
> We have the following results on magnitude pruning which we shall include in the revision.
>
>
> | Model | \% Pruned | Algorithm | Dataset | Baseline | Pruned | Diff |
> |-------|----------|-----------|---------|----------|--------|---|
> | ResNet-20 (200k) | 70 | Magnitude Pruning (IMP) | CIFAR-10 | 92.4 | 91.2 | 1.2 |
> | VGG (16M) | 95 | Magnitude Pruning | CIFAR-10 | 92.40 | 90.20 | 2.2 |
> | VGG (16M) | 98 | Magnitude Pruning | CIFAR-10 | 92.40 | 90.04 | 2.36 |
> | VGG (16M) | 95 | Magnitude Pruning | Tiny-ImageNet | 48.74 | 46.79 | 1.95 |
> | VGG (16M) | 98 | Magnitude Pruning | Tiny-ImageNet | 48.74 | 44.26 | 4.48 |
>
>
> [1] [arXiv:2106.10404](https://arxiv.org/pdf/2106.10404)
>
> [2] [arXiv:2404.03687v1](https://arxiv.org/pdf/2404.03687v1)
>
> **4. Clarifications about Tables 1 and 2**
>
> We apologize for any confusion regarding our table organization. The terms one-shot in table 1 and iterative in table 2 refer to whether causal pruning is single shot or iterative.
>
> - *Table 1 (One-shot)*: Compares our one-shot causal pruning against CHITA, a recent state-of-the-art method
> - *Table 2 (Iterative)*: Compares our iterative approach against various established methods to demonstrate competitive performance across different algorithmic families
>
> The inclusion of methods like SNIP in table 2 reflects their strong empirical performance rather than methodological similarity.
>
> Due to computational constraints, we report results at the sparsity levels from original papers rather than standardized percentages, which explains the variation in sparsity levels across methods. Nevertheless, our results demonstrate that causal pruning achieves competitive performance, with lower accuracy drops at comparable or higher pruning percentages.
>
> While we acknowledge that comparing structured (HRank) and unstructured methods has limitations, this comparison provides a general perspective on our method's relative performance across different pruning paradigms.
>
> We shall include all these details in the revised version.
>
> **5. Computational Overhead**
>
> We provide complexity analysis on page 5 (Computational Complexity paragraph). The overhead involves storing SGD trajectories and solving Lasso regression, which scales reasonably with network size. Specific timing depends heavily on hardware and communication costs. Hence, we have not given these numbers.

---

> ### Author Response · Authors · 2025-08-16
>
> **6. Scope and Additional Experiments (CIFAR100)**
>
> We do understand the requirement for CIFAR100 to check if our approach holds for larger number of classes. However, we include Tiny-ImageNet which has 200 classes. Given our aim to demonstrate the core theoretical insight rather than comprehensive benchmarking, we believe current experiments provide sufficient evidence. We also believe our experimental set up and results conforms to the theoretical claims made in the manuscript and are aligned with the scientific mission of TMLR.
>
>
> To summarize, Our work demonstrates that SGD optimization trajectories contain valuable information about parameter importance via causal relationships. The model collapse experiment when pruning high $\gamma_k$ parameters provides further compelling evidence for this connection. The experiments (sec 5.3) show that the results, although not state-of-the-art, are competitive -- validating our theoretical insight.

---

> ### Comment · Reviewer_Hx2Z · 2025-09-08
>
> I appreciate the addition of the high y_k vs low y_k experiment (Section 5.3) - this provides compelling evidence for your core claim. However, it lacks 1) error bars (how many seeds?) 2) incremental details (did you test 1% each run - 30 data points, or 1%, 5%, 10% etc?).
>
> Additionally, I still have concerns about:
>
> 1. "Remark: Granger causality requires stationarity, which may not hold throughout entire training runs. To
> address this limitation, we employ short training windows with periodic weight resets for data collection – a
> standard practice in the pruning literature." This isn't clear - are you resetting to a recent checkpoint? How does collecting multiple trajectories from the same checkpoint address the fundamental non-stationarity of neural network training dynamics?
>
> 2. Table 2 still remains the biggest handicap to this experment as it compares methods at different sparsity levels and with different baselines. For instance, if your baseline model for tiny-imagenet is 48.74 for magnitude, but 56.01 for CP, the delta does not really tell me anything. Also if the smaller delta the better, why is 5.97 (bigger drop off) better than 4.48 according to the highlight?

---

> ### Author Response · Authors · 2025-09-08
>
> **Error bars and incremental details:** We used 3 seeds for all experiments. The updated figure now includes error bars. The x-axis represents sparsity levels tested at 1%, 3%, 5%, 10%, 15%, 20%, 25%, and 30%.
>
> 1a. **are you resetting to a recent checkpoint?** Yes, we reset to recent checkpoints using weight rewinding. Specifically, after pruning in each cycle, weights are reset to their values at the cycle's start while preserving the updated pruning masks.
>
> 1b. **How does collecting multiple trajectories from the same checkpoint address the fundamental non-stationarity of neural network training dynamics?** To clarify: we analyze single trajectories from individual checkpoints within short windows (one pruning cycle), not multiple trajectories from the same checkpoint. This approach mitigates non-stationarity by focusing on locally stationary segments rather than entire training runs
>
> 2a. We appreciate this important clarification. Table 2 only shows that causal pruning achieves competitive performance across different datasets and architectures, not state-of-the-art results. The varying baselines are due to different experimental setups from prior work. The performance drops show that our method maintains reasonable accuracy when using our novel approach. Our primary contribution is the insight connecting SGD trajectories to weight importance, with Table 2 validating that this approach is practically viable.
>
> *Edit:* Although the accuracies themselves might not say much, the drops can still be compared. The drop for causal pruning at 95% is 1.7 while for magnitude is 1.95, and at 98% pruning causal pruning results in a drop of 5.97 vs 4.48 for magnitude. This validates causal pruning approach. We agree (and also do not claim) that this does not imply state-of-the-art.
>
> 2b. We apologise for highlighting the wrong cell. This is corrected in the revised version.

---

### Review · Reviewer_2LSr · 2025-09-03

**Summary Of Contributions:**

This paper introduces Causal Pruning, a novel neural network pruning method that leverages the optimization trajectory of SGD to identify parameter importance. The key insight is to treat each gradient update as a causal intervention and measure the discrepancy between the predicted loss changes via a first-order Taylor approximation and the observed loss changes.

Main contributions/strengths:
1. The paper shows that in early optimization, γ_k captures how well noisy SGD aligns with true gradient descent. At convergence, pruning parameters with γ_k ≈ 0 preserves Hessian flatness. This causality framework for pruning is original.
2. The empirical demonstration that causal pruning outperforms magnitude pruning and leads to flatter minima is solid.

Weakness:
1. This method seems to be impractical if the training trajectory is not available, which is the most cases for a post-training method.
2. While the intuitions are compelling, the theoretical analysis relies heavily on first-order Taylor approximations and makes simplifying assumptions (e.g., constant Hessian in second-order analysis) that may not hold in practice.

**Audience:**

No

**Audience Explanation:**

This paper lacks empirical interest: it only prunes small networks on vision tasks and fails to demonstrate significant improvements. Additionally, it appears highly impractical as a post-training method. The paper is also theoretically uncompelling, as it lacks sufficient lemmas and proofs to establish the connection between training trajectory and parameter importance.

**Broader Impact Concerns:**

No ethical issue detected.

**Claims And Evidence:**

No

**Claims Explanation:**

Empirically, if the goal of using the trajectory is to estimate the influence of parameters on loss, and leads to flat local minima, why don't you compare with the Hessian-aware pruning method [1,2]?


Some theoretical claims lack rigorous proofs:
1. The claim that γ_k = 0 implies parameters can be pruned "without significantly affecting the Hessian-based variation" (Section 4.2) is stated without formal proof.
2. The extension to momentum-based optimizers (Appendix A) is sketchy and acknowledges difficulties with adaptive learning rates.
3. The synthetic Friedman example (Figure 1), while illustrative, doesn't strongly validate the method for neural networks.

[1] Global Vision Transformer Pruning with Hessian-Aware Saliency
[2] Hessian-Aware Pruning and Optimal Neural Implant

**Requested Changes:**

1. Could you provide the detailed derivation for equation 12? The current presentation jumps from stating the optimization problem to the solution $\gamma \_\mathrm{k}=\left(\Sigma \_\mathrm{t}\right. \left.\nabla \_k L\left(\theta^{\wedge} t\right) \Delta \theta \_k^{\wedge} t\right) /\left(\Sigma \_t\left(\Delta \theta \_k^{\wedge} t\right)^2\right)$. When minimizing $\Sigma \_t\left(\nabla \_k L\left(\theta^{\wedge} t\right)-\gamma \_k \Delta \theta \_k^{\wedge} t\right)^2$ as stated, the standard least squares solution would be different. Could you show the intermediate steps or clarify if a different objective is being minimized?

2. The paper claims gradient descent has "implicit causality" with a uniform influence coefficient $-1 / \eta$ for all parameters. However, Granger causality typically requires comparing restricted vs unrestricted models with statistical significance testing. Could you elaborate on how your formulation satisfies the formal requirements of Granger causality, particularly given the model form mismatch mentioned in Concern 1?

---

> ### Author Response · Authors · 2025-09-06
>
> We thank the reviewer for their detailed feedback and for acknowledging that our "causality framework for pruning is original" and that our "empirical demonstration of flatness is solid."
>
> We respectfully disagree with the assessment that our claims lack supporting evidence. The combination of novel framework, mathematical derivations, flatness preservation experiments, specifically the model collapse validation provides substantial support for our core claims.
>
> **1. Core Contribution and Scope**
>
> Before addressing specific concerns, we clarify that our primary contribution is demonstrating a surprising connection between parameter importance and causal information embedded in SGD trajectories. To our knowledge, this fundamental insight has not been explored before. While we do not claim state-of-the-art performance, our work opens a new perspective on understanding parameter importance through optimization dynamics.
>
> **2. Practicality and Post-Training Application**
>
> Our method can be adapted for post-training scenarios by starting from pre-trained checkpoints and collecting trajectory data through short additional training runs. We only require parameter updates and loss values, making the data collection lightweight. The procedure is scalable in principle, as demonstrated by our ResNet-50/ImageNet experiments.
>
> *First-Order Approximations:* The principle underlying Taylor expansion is that when first-order effects predominate, then a first-order approximation is reasonable. Our empirical results, particularly the model collapse experiment, suggest this approximation captures meaningful relationships in practice.
>
> **3. Mathematical Derivation for Equation 12**
> We provide the detailed derivation requested:
>
> \begin{equation}
>     \gamma_k^* = \arg \min_{\gamma_k} \sum_{t=1}^T (\nabla_k L(\theta^t) - \gamma_k \Delta\theta_k^t)^2
> \end{equation}
> Taking the derivative with respect to $\gamma_k$,
> \begin{equation}
>     \sum_{t=1}^T \frac{\partial}{\partial \gamma_k} (\nabla_k L(\theta^t) - \gamma_k \Delta\theta_k^t)^2
> \end{equation}
> \begin{equation}
>     = -2 \sum_{t=1}^T (\nabla_k L(\theta^t) - \gamma_k \Delta\theta_k^t) \Delta\theta_k^t
> \end{equation}
> Setting this equal to $0$ and solving for $\gamma_k$, we get
> \begin{equation}
>     \gamma_k^* = \frac{\sum_{t=1}^T \nabla_k L(\theta^t) \Delta\theta_k^t}{\sum_{t=1}^T (\Delta\theta_k^t)^2}
> \end{equation}
>
> **Theoretical Claims and Hessian Analysis**
>
> *Regarding the $\gamma_k$ = 0 claim*: Observe that equation 15 can be written as $\gamma^* = \min E[(x^t H x - x^t \text{diag}(\gamma) x)^2]$ which is equal to $\min E[(x^t (H - \text{diag}(\gamma)) x)^2]$. So, the optimal $\gamma$ is the best diagonal approximation of $H$ when the ``data'' comes from $x$ (which are the gradient around local minima). If we have that the $k^{th}$ component of $\gamma^*$ is small, then this indicates that parameter contributes negligibly to the second-order term $x^T Hx$. Consequently, removing this parameter should not significantly affect Hessian-based variation in the loss landscape.
>
> Note that $x$ depends on the dataset and hence more formal statements are not possible unless assumptions are made. For example, if we further assume that $E(x_i x_j) = 0$, then eq(15) gives that $\gamma_{k} = H_{kk}$. Hence if $(\gamma_k)^* \approx 0$, $H_{kk} \approx 0$ and hence the trace of hessian remains the same.
>
> While we acknowledge this reasoning could be formalized further, for the purpose of this article, we believe the intuition from equation 15, and the empirical verification demonstrating preserved flatness after pruning are sufficient to justify our claims.
>
> *Comparing with other hessian based approaches:* We appreciate the references to Hessian-aware pruning methods. Please note that our primary baseline CHITA is also an hessian aware technique (approximates hessian efficiently). Please also note that -- Our contribution lies in this novel perspective rather than direct competition with existing methods.
>
> *Critical Validation Experiment:* Following the reviewer Hx2Z suggestion about empirical validation, we conducted a crucial experiment: pruning high $\gamma$ parameters causes catastrophic model collapse. On ResNet-20/CIFAR-10, pruning just 30\% of high $\gamma$ parameters reduced accuracy to ~10\%, while pruning the same percentage of low $\gamma$ parameters maintains original performance. This strongly supports our claim that $\gamma_k$ coefficients capture functional importance.
>
> **Clarification of Granger Causality Framework**
> We use Granger causality as an intuitive framework rather than formal statistical testing. Our approach is computationally cheaper while capturing the essential idea. Comparing equations (4)-(6) with (8)-(9), we see that standard gradient descent assumes $\gamma_k = -1/\eta$ uniformly, which we interpret as *implicit causality* with uniform influence coefficients. Our method learns parameter-specific coefficients, providing more nuanced importance estimates.

---

> > ### Author Response · Authors · 2025-09-06
> >
> > **Experimental Scope**
> > Please note that our experiments include ResNet-50/ImageNet. Given our focus on demonstrating fundamental principles rather than comprehensive benchmarking, current experiments adequately support our insights. The architecture-agnostic nature of our approach suggests broader applicability as future work.
> >
> > Our work demonstrates a fundamentally novel and empirically validated connection between SGD optimization trajectories and parameter importance. The model collapse experiment, flatness preservation results, and competitive pruning performance provide compelling evidence for this relationship. We believe this opens valuable new directions for understanding the optimization-pruning connection and contributes meaningfully to the foundations of neural network compression.

---

### Author Response · Authors · 2025-09-06

We thank the AE and the reviewers for their detailed comments and in improving the quality of the article.

We have uploaded a revision with the following changes:

1. Thanks to the reviewer's comments, we included the additional experiment removing the top scores instead of the least scores for both causal and magnitude pruning. We hope to see that in this reverse scenario, good heuristics suffer more degradation. And we show that causal pruning does suffer more degradation. This is described in (new) section 5.3.

*Note:* In our earlier experiments (comments) we observed a catastrophic collapse. However, we later realised that this was because the final classification layer was pruned. We corrected this by ignoring the classification layer when pruning and by comparing with the magnitude baseline.

2. Apart from above we have also improved the explanation at several places:

- We add the following statement to address stationarity concerns

```
Granger causality requires stationarity, which may not hold throughout entire training runs. To address this limitation, we employ short training windows with periodic weight resets for data collection – a standard practice in the pruning literature.
```

- We add the following to motivate the design of our experiment in section 5.2

```
To assess the sharpness of minima induced by causal pruning, we evaluate on CIFAR-10 using two architectures of contrasting complexity – LeNet and ResNet18. This choice is motivated by the principle that demonstrating flatter minima at architectural extremes provides compelling evidence for the general flatness property.
```

- We clarify the categorisation of baselines in tables (1) and (2)

- We also included magnitude pruning and iterative magnitude pruning results.

---

### Comment · Action_Editor_sr7r · 2025-10-07
**Reviewers should give their official recommendation**

Dear reviewers,

Thank you for your time and effort in reviewing this work. If you haven't already, please submit your official recommendation. If you have any questions, don't hesitate to reach out.

Best,

AE

---

### Decision · Action_Editor_sr7r · 2025-10-09

**Recommendation:** Reject

**Additional Comments:**

I am unfortunately recommending reject given the reviewers concerns about empirical validation. I believe this is an addressable problem (namely, do experiments that are "standard" in the sparsity field - maybe use the ShrinkBench framework?https://github.com/JJGO/shrinkbench), but given that TMLR does not allow for major revisions (or another round of reviews), I have to reject at this time. The authors may consider submitting again (after addressing the concerns the reviewers made). Alternatively, perhaps submitting to a conference more aligned with sparsity (e.g. CPAL - conference on parsimony and learning) may be found to be a good alternative route. Either way, I think this work has merit but also can be strengthened.

**Audience:**

Yes

**Audience Explanation:**

New ideas on sparsifying neural networks, especially from different perspectives, is something of interest to the TMLR community.

**Claims And Evidence:**

No

**Claims Explanation:**

All three reviewers agree that the claims we not supported by clear and convincing evidence. The primary concern by reviewers was that the "empirical validation lacks breadth and standardization (limited architectures, inconsistent baselines, and sparse benchmarking)". TMLR submissions certainly do not have to achieve state-of-the-art or have the most detailed benchmarking, but given that the opinion that the empirical validation needed to be strengthened was an opinion shared by all 3 reviewers, I unfortunately have to conclude that the claims we not supported by clear and convincing evidence.

**Resubmission Of Major Revision:**

The authors may consider submitting a major revision at a later time.